# DENSE RGB SLAM WITH NEURAL IMPLICIT MAPS

**Heng Li**[123]*, **Xiaodong Gu**[2]*, **Weihao Yuan**[2], **Luwei Yang**[3], **Zilong Dong**[2], **Ping Tan**[123]
[1]Hong Kong University of Science and Technology,
[2]Alibaba Group, [3]Simon Fraser University
lh.heng.li@connect.ust.hk, luweiy@sfu.ca, pingtan@ust.hk
{qianmu.ywh, dadong.gxd, list.dzl}@alibaba-inc.com

## ABSTRACT

There is an emerging trend of using neural implicit functions for map representation in Simultaneous Localization and Mapping (SLAM). Some pioneer works have achieved encouraging results on RGB-D SLAM. In this paper, we present a dense RGB SLAM method with neural implicit map representation. To reach this challenging goal without depth input, we introduce a hierarchical feature volume to facilitate the implicit map decoder. This design effectively fuses shape cues across different scales to facilitate map reconstruction. Our method simultaneously solves the camera motion and the neural implicit map by matching the rendered and input video frames. To facilitate optimization, we further propose a photometric warping loss in the spirit of multi-view stereo to better constrain the camera pose and scene geometry. We evaluate our method on commonly used benchmarks and compare it with modern RGB and RGB-D SLAM systems. Our method achieves favorable results than previous methods and even surpasses some recent RGB-D SLAM methods. The code is at poptree.github.io/DIM-SLAM/.

## 1 INTRODUCTION

Visual SLAM is a fundamental task in 3D computer vision with many applications in AR/VR and robotics. The goal of visual SLAM is to estimate the camera poses and build a 3D map of the environment simultaneously from visual inputs. Visual SLAM methods can be primarily divided into sparse or dense according to their reconstructed 3D maps. Sparse methods (Mur-Artal & Tardós, 2017; Engel et al., 2017) focus on recovering camera motion with a set of sparse or semi-dense 3D points. Dense works (Newcombe et al., 2011b) seek to recover the depth of every observed pixel and are often more desirable for many downstream applications such as occlusion in AR/VR or obstacle detection in robotics. Earlier methods (Newcombe et al., 2011a; Whelan et al., 2012) often resort to RGB-D cameras for dense map reconstruction. However, RGB-D cameras are more suitable to indoor scenes and more expensive because of the specialized sensors.

Another important problem in visual SLAM is map representation. Sparse SLAM methods (Mur-Artal & Tardós, 2017; Engel et al., 2017) typically use point clouds for map representation, while dense methods (Newcombe et al., 2011b;a) usually adopt triangle meshes. As observed in many recent geometry processing works (Mescheder et al., 2019; Park et al., 2019; Chen & Zhang, 2019), neural implicit function offers a promising presentation for 3D data processing. The pioneer work, iMAP (Sucar et al., 2021), introduces an implicit map representation for dense visual SLAM. This map representation is more compact, continuous, and allowing for prediction of unobserved areas, which could potentially benefit applications like path planning (Shrestha et al., 2019) and object manipulation (Sucar et al., 2020). However, as observed in NICE-SLAM (Zhu et al., 2022), iMAP (Sucar et al., 2021) is limited to room-scale scenes due to the restricted representation power of MLPs. NICE-SLAM (Zhu et al., 2022) introduces a hierarchical feature volume to facilitate the map reconstruction and generalize the implicit map to larger scenes. However, both iMAP (Sucar et al., 2021) and NICE-SLAM (Zhu et al., 2022) are limited to RGB-D cameras.

This paper presents a novel dense visual SLAM method with regular RGB cameras based on the implicit map representation. We also adopt a hierarchical feature volume like NICE-SLAM to deal

---

*Equal contribution

with larger scenes. But our formulation is more suitable for visual SLAM. Firstly, the decoders in NICE-SLAM (Zhu et al., 2022) are pretrained, which might cause problems when generalizing to different scenes (Yang et al., 2021), while our method learns the scene features and decoders together on the fly to avoid generalization problem Secondly, NICE-SLAM (Zhu et al., 2022) computes the occupancy at each point from features at different scales respectively and then sums these occupancies together, while we fuse features from all scales to compute the occupancy at once. In this way, our optimization becomes much faster and thus can afford to use more pixels and iterations, enabling our framework to work on the RGB setting. In experiments, we find the number of feature hierarchy is important to enhance the system accuracy and robustness. Intuitively, features from fine volumes capture geometry details, while features from coarse volumes enforce geometry regularity like smoothness or planarity. While NICE-SLAM (Zhu et al., 2022) only optimizes two feature volumes with voxel sizes of 32cm and 16cm, our method solves six feature volumes from 8cm to 64cm. Our fusion of features across many different scales leads to more robust and accurate tracking and mapping as demonstrated in experiments.

Another challenge in our setting is that there are no input depth observations. Therefore, we design a sophisticated warping loss to further constrain the camera motion and scene map in the same spirit of multi-view stereo (Zheng et al., 2014; Newcombe et al., 2011b; Wang et al., 2021b; Yu et al., 2021). Specifically, we warp one frame to other nearby frames according to the estimated scene map and camera poses and optimize the solution to minimize the warping loss. However, this warping loss is subject to view-dependent intensity changes such as specular reflections. To address this problem, we carefully sample image pixels visible in multiple video frames and evaluate the structural similarity of their surrounding patches to build a robust system.

We perform extensive evaluations on three different datasets and achieve state-of-the-art performance on both mapping and camera tracking. Our method even surpasses recent RGB-D based methods like iMAP (Sucar et al., 2021) and NICE-SLAM (Zhu et al., 2022) on camera tracking.

Our contributions can be summarized as the following:

• We design the first dense RGB SLAM with neural implicit map representation,

• We introduce a hierarchical feature volume for better occupancy evaluation and a multiscale patch-based warping loss to boost system performance with only RGB inputs,

• We achieve strong results on benchmark datasets and even surpass some recent RGB-D methods.

## 2 RELATED WORK

**Visual SLAM** Many visual SLAM algorithms and systems have been developed the last two decades. We only quickly review some of the most relevant works, and more comprehensive surveys can be found at (Cadena et al., 2016; Macario Barros et al., 2022). Sparse visual SLAM algorithms (Klein & Murray, 2007; Mur-Artal & Tardós, 2017) focus on solving accurate camera poses and only recover a sparse set of 3D landmarks serving for camera tracking. Semi-dense methods like LSD-SLAM (Engel et al., 2014) and DSO (Engel et al., 2017) achieve more robust tracking in textureless scenes by reconstructing the semi-dense pixels with strong image gradients. In comparison, dense visual SLAM algorithms (Newcombe et al., 2011b) aim to solve the depth of every observed pixel, which is very challenging, especially in featureless regions.

In the past, dense visual SLAM is often solved with RGB-D sensors. KinectFusion (Newcombe et al., 2011a) and the follow-up works (Whelan et al., 2012; 2021; Dai et al., 2017) register the input sequence of depth images through Truncated Signed Distance Functions (TSDFs) to track camera motion and recover a clean scene model at the same time. Most recently, iMAP and NICE-SLAM, introduce neural implicit functions as the map presentation and achieve better scene completeness, especially for unobserved regions. All these methods are limited to RGB-D cameras.

More recently, deep neural networks are employed to solve dense depth from regular RGB cameras. Earlier methods (Zhou et al., 2017; Ummenhofer et al., 2017; Zhou et al., 2018) directly regress camera ego-motion and scene depth from input images. Later, multi-view geometry constraints are enforced in (Bloesch et al., 2018; Tang & Tan, 2019; Teed & Deng, 2020; Wei et al., 2020; Teed & Deng, 2021) for better generalization and accuracy. But they all only recover depth maps instead of complete scene models. Our method also employs deep neural networks to solve the challenging

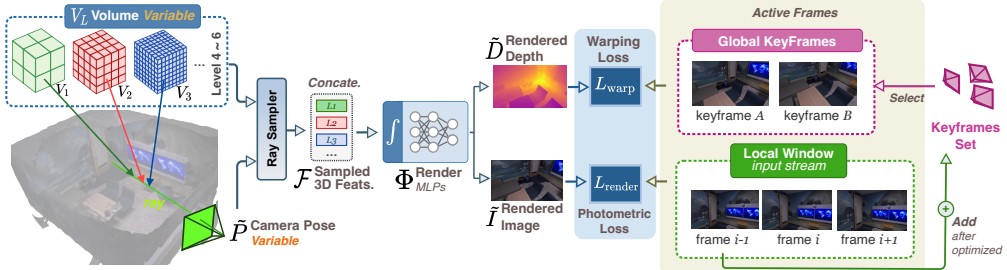

Figure 1: Pipeline of our framework. Given a camera pose, our method sample the multi-scale features along view rays and pass the concatenated features through an MLP decoder to compute the depth and color at each pixel. In this way, we can solve the camera motion and a 3D scene map (captured by the feature volume and MLP decoder) simultaneously by matching the rendered images with observed ones. We further include warping loss across different views to enhance the system performance.

dense visual SLAM problem. Unlike these previous methods that only recover a depth map per frame, we follow iMAP (Sucar et al., 2021) and NICE-SLAM (Zhu et al., 2022) to compute a neural implicit function for map representation.

**Neural Implicit Representations** Neural implicit functions have been widely adopted for many different tasks in recent years. It has been introduced for 3D shape modeling (Mescheder et al., 2019; Park et al., 2019; Chen & Zhang, 2019), novel view synthesis (Mildenhall et al., 2020; Zhang et al., 2020; Martin-Brualla et al., 2021; Barron et al., 2021; Sara Fridovich-Keil and Alex Yu et al., 2022; Müller et al., 2022), clothed human reconstruction (Saito et al., 2019; 2020), scene reconstruction (Murez et al., 2020; Sun et al., 2021), and object reconstruction (Yariv et al., 2020; Niemeyer et al., 2020; Wang et al., 2021a), etc. But all these works typically require precisely known camera poses. There are only a few works (Jeong et al., 2021; Zhang et al., 2022; Lin et al., 2021) trying to deal with uncalibrated cameras. Furthermore, all these methods require a long optimization process, and hence, are unsuitable for real-time applications like visual SLAM.

Recently, the two pioneer works, iMAP (Sucar et al., 2021) and NICE-SLAM (Zhu et al., 2022), both adopt neural implicit functions to represent the scene map in visual SLAM. These works can solve camera poses and 3D scene maps in real-time by enforcing the rendered depth and color images to match with the input ones. However, they both require RGB-D sequences as input, which requires specialized sensors. In comparison, our visual SLAM method works with regular RGB cameras and solves camera poses and the 3D scene simultaneously in real-time.

## 3 APPROACH

The overall pipeline of our framework is shown in Figure 1. Given an RGB video as input, our method aims to recover a 3D scene map and the camera motions simultaneously. We represent the scene map by a neural implicit function with a learnable multi-resolution feature volume. By sampling features from the volume grid along view rays and querying the sampled feature with the MLP decoder, we can render the depth and color of each pixel given the estimated camera parameters. Since this rendering process is differentiable, we can simultaneously optimize the neural implicit map as well as the camera poses by minimizing an objective function defined on photometric rendering loss and warping loss.

### 3.1 IMPLICIT MAP REPRESENTATION

This section introduces the formulation of our implicit map representation that combines a learnable multi-resolution volume encoding $\{V_l\}$ and an MLP decoder $\Phi$ for depth and color prediction.

**Multi-resolution Volume Encoding** Directly representing the scene map with MLPs, which maps a 3D point to its occupancy and color, confronts a forgetting problem because the MLP is globally updated for any frame (Sucar et al., 2021). To address this, we equip the MLP with multi-resolution volumes $\{V_l\}_{l=1}^{L}$, which are updated locally on seen regions of each frame(Sara Fridovich-Keil and Alex Yu et al., 2022; Müller et al., 2022). The input point is encoded by the feature $\mathcal{F}$ sampled from the volumes $\{V_l\}$, which could also explicitly store the geometric information. We adopt a combination of $L$ volumes, whose resolution arrange from the voxel size of $v_{max}$ to the voxel size of

$v_{min}$. This hierarchical structure works better than a single-scale volume because the gradient-based camera optimization on high-resolution volume is susceptible to a suboptimal solution if without a good initialization. In contrast, in a coarse-to-fine architecture, the low-resolution volume could enforce the smoothness in 3D space in early registration, while the high-resolution volume could encode the shape details. To compute this encoding of a given 3D point $\mathbf{p} = (x, y, z)$, we first interpolate feature $V_l(\mathbf{p})$ at the point $\mathbf{p}$ by trilinear sampling. Then the 3D feature encoding $\mathcal{F}(\mathbf{p})$ is obtained by concatenating the features from all levels. Notably, the feature channel of each volume is set to 1. The small feature channel reduces memory consumption without losing performance, which is demonstrated in the ablation study.

**Color and Depth Prediction** The MLP decoder $\Phi$ consists of three hidden layers with a feature channel of 32 and two heads that output the color $\mathbf{c_p}$ and density $o_\mathbf{p}$ as,

$$(o_\mathbf{p}, \mathbf{c_p}) = \Phi(\mathcal{F}(\mathbf{p})). \tag{1}$$

However, extracting scene geometry from the computed density requires careful tuning of the density threshold and leads to artifacts due to the ambiguity present in the density field. To address this problem, following (Oechsle et al., 2021), a sigmoid function is added on the output of $o_\mathbf{p}$ to regularize it to $[0, 1]$, in which case a fixed level set of 0.5 could be used to extract the mesh. Also, the ray termination probability of point $\mathbf{p}_i$ is now computed as,

$$w_i = o_{p_i} \prod_{j=0}^{i-1} (1 - o_{p_j}). \tag{2}$$

During volume rendering, a hierarchical sampling strategy similar to NeuS (Wang et al., 2021a) is adopted. We first uniformly sample 32 points on a view ray and then subsequently conducts importance sampling of additional 16 points for 4 times based on the previously estimated weights. $N = 96$ points are sampled in total. For each point, we compute its depth $\tilde{D}$ and color $\tilde{\mathbf{I}}$ as,

$$\tilde{D} = \sum_{i=0}^{N} w_i z_i, \tilde{\mathbf{I}} = \sum_{i=0}^{N} w_i \mathbf{c}_i, \tag{3}$$

where $z_i$ and $\mathbf{c}_i$ are the depth and color of the point $\mathbf{p}_i$ along the viewing ray. Notice that since the rendering equation is differentiable, the camera pose, multi-resolution volume encoding, and the parameters of the MLP decoder could be optimized together from the gradient back-propagation.

## 3.2 JOINT OPTIMIZATION

In this section, we present the objective function for optimization. The implicit scene representation and the camera poses are jointly optimized in a set of frames $\mathcal{W}$ with corresponding poses $\tilde{\mathcal{P}} : \{\tilde{\mathbf{P}}_k = [\tilde{\mathbf{R}}_k, \tilde{\mathbf{t}}_k], k \in \mathcal{W}\}$, where $\mathbf{R}$ and $\mathbf{t}$ are the rotation and translation camera parameters respectively. The frame selection strategy is explained later in section 3.3.

**Photometric Rendering Loss** To optimize the scene map and the camera poses, we compute the L1 loss between the rendered and observed color for a set of randomly sampled pixels $\mathcal{M}$:

$$L_{\text{render}} = \frac{1}{|\mathcal{M}|} \sum_{\mathbf{q} \in \mathcal{M}} ||\mathbf{I_q} - \tilde{\mathbf{I}_q}||_1. \tag{4}$$

Here, $|\mathcal{M}|$ denotes the number of sampled pixels. This rendering loss alone is insufficient to determine the scene geometry and could lead to ambiguous solutions, where the rendered image matches the observation well but the depth estimation is completely wrong. To overcome this problem, we introduce a photometric warping loss to further constrain the optimization problem.

**Photometric Warping Loss** We define the photometric warping loss in a similar spirit as multi-view stereo (Zheng et al., 2014) to enforce the geometry consistency. Let $\mathbf{q}_k$ denotes a 2D pixel in frame $k$, we first lift it into 3D space and then project it to another frame $l$ as:

$$\mathbf{q}_{k \to l} = \mathbf{K}_l \tilde{\mathbf{R}}_l^\top (\tilde{\mathbf{R}}_k \mathbf{K}_k^{-1} \mathbf{q}_k^{\text{homo}} \tilde{D}_{\mathbf{q}_k} + \tilde{\mathbf{t}}_k - \tilde{\mathbf{t}}_l), \tag{5}$$

where $\mathbf{q}_k^{\text{homo}} = (u, v, 1)$ is the homogeneous coordinate of $\mathbf{q}_k$, and $\tilde{D}_{\mathbf{q}_k}$ denotes its estimated depth. We minimize the warping loss defined as the following,

$$L_{\text{warping}} = \frac{1}{|\mathcal{M}|} \sum_{\mathbf{q}_k \in \mathcal{M}} \sum_{l \in \mathcal{W}, k \neq l} ||\mathbf{I}_{\mathbf{q}_k} - \mathbf{I}_{\mathbf{q}_{k \to l}}||_1. \tag{6}$$

However, minimizing the warping loss on a single pixel does not model the view-dependent intensity changes, e.g., specularities, and could create artifacts in the scene map. We thus define the warping loss over image patches instead of individual pixels to address this issue. For an image patch $\mathcal{N}^s_{\mathbf{q}_k}$ centered on pixel $\mathbf{q}_k$ and with the size of $s \times s$, we assume all pixels on the patch have the same depth value as $\mathbf{q}_k$. Similar to the case of a single pixel, we project all pixels in this patch to another frame $l$ and obtain $\mathcal{N}^s_{\mathbf{q}_{k \to l}}$. Thus, we extend the pixel-wise warping loss to patch-wise as,

$$L_{\text{warping}} = \frac{1}{|\mathcal{M}|} \sum_{\mathbf{q}_k \in \mathcal{M}} \sum_{l \in \mathcal{W}, k \neq l} \sum_{s \in \mathcal{S}} B_{\mathbf{q}_k} \text{SSIM}(\mathcal{N}^s_{\mathbf{q}_k}, \mathcal{N}^s_{\mathbf{q}_{k \to l}}), \tag{7}$$

where $B_{\mathbf{q}_k}$ denotes a visibility mask which will be explained later, and $\mathcal{S} = \{1, 5, 11\}$ denotes the patch size. In practice, we select the structure similarity loss (SSIM) (Wang et al., 2004) as it achieves better results than other robust functions, e.g. (Park et al., 2017). Note that this formulation degenerates pixel-wise when a patch of size 1 is used. Further, this warping assumes front-parallel image patches, which might be improved with a perspective warping based on surface normals extracted from the scene map. Here, we choose the front-parallel warping for its better computation efficiency.

**Visibility Mask** We define a binary visibility mask $B_{\mathbf{q}_k}$ to filter pixels with low visibility. For any pixel $\mathbf{q}_k$, we count the number of its visible frames by projecting it to all other frames in the set $\mathcal{W}$. The projection is counted as valid if the projected pixel is within the frame boundary. Only pixels with more than 5 valid projections is regarded as visible, whose visibility value is set to 1. Note that we do not model occlusion here for better efficiency.

**Regularization Loss** To deal with textureless regions, we enforce an edge-preserving regularization term to smooth the rendered depth $\tilde{D}$ as,

$$L_{\text{smooth}} = \sum_{k \in \mathcal{W}} e^{-||\nabla \tilde{\mathbf{I}}||_2} ||\nabla \tilde{D}||_1, \tag{8}$$

where $\nabla \tilde{\mathbf{I}}, \nabla \tilde{D}$ are the first-order gradients of the computed color and depth images, respectively. Thus, the final objective for mapping and tracking in a single thread is obtained as,

$$\underset{\Theta, \mathcal{P}_k, k \in \mathcal{W}}{\arg \min} \ \alpha_{\text{warping}} L_{\text{warping}} + \alpha_{\text{render}} L_{\text{render}} + \alpha_{\text{smooth}} L_{\text{smooth}}, \tag{9}$$

where $\alpha$ denotes the weighting factors, and $\Theta$ denotes the parameters of the implicit scene representation, including the multi-resolution volumes and the MLP decoder.

## 3.3 System

In this section, we introduce several crucial components to build our complete visual SLAM system.

**Initialization** The initialization is performed when a small set of frames (13 frames in all our experiments) are collected. The pose of the first frame is set to the identity matrix. Then the poses of the remaining frames are optimized from the objective function for $N_i$ iterations. After the initialization, the first frame is added to the global keyframe set and fixed. The parameters of MLP decoders are also fixed after initialization.

**Window Optimization** During camera tracking, our method always maintains a window of active frames (21 frames in all our experiments). The active frames include two different types: local window frames and global keyframes. For a frame $k$, we regard the frames with index from $k - 5$ to $k + 5$ as the local window frames. We further sample 10 keyframes from the global keyframe set. Specifically, we randomly draw 100 pixels on frame $k$ and project them to all keyframes using estimated camera poses $\tilde{\mathbf{P}}$ and depths $\tilde{D}$. We then evaluate the view overlap ratio by counting the percentage of the pixels projected inside the boundary of those keyframes. Afterward, we randomly select 10 keyframes from those with an overlap ratio over $70\%$ to form the global keyframe set. Then, all these active frames are used to optimize the implicit scene representation and the poses for $N_w$ iterations. After the window optimization, the oldest local frame $k - 5$ is removed, and a new incoming frame of $k + 6$ with a constant velocity motion model is added to the set of local window frames. The camera poses of all global keyframes are fixed during tracking. Please refer to the appendix about windows optimization for mapping in two threads cases.

**Keyframe Selection** We follow the simple keyframe selection mechanism similar to iMAP and NICE-SLAM, while a more sophisticated method might be designed based on image retrieval and

|  |  | o-0 | o-1 | o-2 | o-3 | o-4 | r-0 | r-1 | r-2 | Avg. |
|---|---|---|---|---|---|---|---|---|---|---|
| NICE-SLAM |  | 1.02 | 3.91 | 1.76 | 6.10 | 11.9 | 4.37 | 2.76 | 3.54 | 4.4 |
| ORB-SLAM2(RGB) |  | 0.43 | **0.30** | 12.2 | 0.39 | 11.4 | **0.30** | 0.42 | **0.25** | 3.21 |
| DROID-SLAM | ATE RMSE[cm]↓ | **0.25** | 0.45 | **0.32** | 0.44 | 0.37 | 0.35 | **0.33** | 0.28 | **0.35** |
| Ours(two threads) |  | 0.89 | 0.72 | 1.11 | 0.75 | 0.92 | 0.84 | 1.22 | 0.66 | 0.89 |
| Ours(one thread) |  | 0.67 | 0.37 | 0.36 | **0.33** | **0.36** | 0.48 | 0.78 | 0.35 | 0.46 |
| iMAP | **Acc.**[cm]↓ | 5.87 | 3.71 | 4.81 | 4.27 | 4.83 | 3.58 | 3.69 | 4.68 | 4.43 |
|  | **Comp.**[cm]↓ | 6.11 | 5.26 | 5.65 | 5.45 | 6.59 | 5.06 | 4.87 | 5.51 | 5.56 |
|  | **Comp. Ratio**[≤ 5cm%]↑ | 77.71 | 79.64 | 77.22 | 77.34 | 77.63 | 83.91 | 83.45 | 75.53 | 79.06 |
| NICE-SLAM | **Acc.**[cm]↓ | **2.26** | 2.50 | 3.82 | 3.50 | 2.77 | 2.73 | **2.58** | **2.65** | **2.85** |
|  | **Comp.**[cm]↓ | **2.02** | 2.36 | **3.57** | **3.83** | 3.84 | 2.87 | 2.47 | 3.00 | **3.00** |
|  | **Comp. Ratio**[≤ 5cm%]↑ | **94.93** | **92.61** | **85.20** | **82.98** | **86.14** | **90.93** | **92.80** | **89.07** | **89.33** |
| DI-Fuison | **Acc.**[cm]↓ | 70.56 | **1.42** | **2.11** | **2.11** | **2.02** | **1.79** | 49.00 | 26.17 | 19.40 |
|  | **Comp.**[cm]↓ | 3.58 | **2.20** | 4.83 | 4.71 | 5.84 | 3.57 | 39.40 | 17.35 | 10.19 |
|  | **Comp. Ratio**[≤ 5cm%]↑ | 87.17 | 91.85 | 80.13 | 79.94 | 80.21 | 87.77 | 32.01 | 45.61 | 72.96 |
| DROID-SLAM | **Acc.**[cm]↓ | 3.26 | 2.92 | 4.90 | **4.47** | 5.35 | 4.01 | 4.53 | **4.40** | 4.23 |
|  | **Comp.**[cm]↓ | 4.22 | 5.54 | 7.97 | 7.77 | 6.80 | 7.10 | 5.72 | 9.43 | 6.82 |
|  | **Comp. Ratio**[≤ 5cm%]↑ | 77.57 | 72.95 | 59.85 | 62.30 | 66.50 | 68.35 | 71.68 | 68.76 | 68.50 |
| Ours(two threads) | **Acc.**[cm]↓ | 2.81 | 2.33 | 5.02 | 5.61 | 4.83 | **3.61** | 4.12 | 6.06 | 4.29 |
|  | **Comp.**[cm]↓ | 2.71 | 3.25 | 6.63 | 6.33 | 6.11 | 5.97 | 5.31 | 7.43 | 5.46 |
|  | **Comp. Ratio**[≤ 5cm%]↑ | 88.56 | 84.7 | 71.40 | 72.54 | 73.23 | 81.12 | 76.94 | 70.85 | 77.41 |
| Ours(one thread) | **Acc.**[cm]↓ | **2.60** | **2.02** | **4.50** | 5.43 | **4.57** | 3.68 | **3.64** | 5.84 | **4.03** |
|  | **Comp.**[cm]↓ | **2.65** | **3.31** | **6.09** | 5.98 | 5.81 | 5.32 | 4.72 | 5.70 | 4.20 |
|  | **Comp. Ratio**[≤ 5cm%]↑ | **89.57** | **84.9** | **75.3** | 73.48 | 76.62 | 82.2 | 80.4 | 74.44 | 79.6 |

Table 1: Quantitative results on the *Replica* dataset. Here, 'o-x' and 'r-x' denote `office-x` and `room-x` sequence respectively. The top part shows the results of tracking, and the lower part shows the results of mapping. DROID-SLAM and our method work with the RGB inputs, while the others are all based on RGB-D data. The best results in RGB and RGB-D SLAMs are **bolded** respectively.

visual localization methods like (Arandjelovic et al., 2016; Brachmann et al., 2017). Specifically, a new keyframe is added if the field of view changes substantially. This view change is measured by the mean square $f$ of the optical flow between the last keyframe and the current frame. A new keyframe is added if $f > T_{kf}$, where $T_{kf} = 10$ pixels in all our experiments.

## 4 EXPERIMENTS

**Datasets** We evaluate our method on three widely used datasets: *TUM RGB-D* (Sturm et al., 2012a), *EuRoC* (Burri et al., 2016), and *Replica* (Straub et al., 2019) dataset. The *TMU RGB-D* and *EuRoC* dataset are captured in small to medium-scale indoor environments. Each scene in the *TUM* dataset contains a video stream from a handheld RGB-D camera associated with the ground-truth trajectory. The *EuRoC* dataset captures stereo images using a drone, and their ground-truth 6D poses are provided with a Vicon System. We use only the monocular setting in the *EuRoC* dataset. The *Replica* dataset contains high-quality 3D scene models. We use the camera trajectories provided by iMAP (Sucar et al., 2021) to render the input RGB sequences.

**Implementation Details** Our method is implemented with PyTorch (Paszke et al., 2017) and runs on a server with two NVIDIA 2080Ti GPUs. Our single-thread implementation, i.e. ours (one thread), and two-thread implementation, i.e. ours (two thread), need one and two GPUs respectively. The voxel sizes of the multi-resolution volumes for ours (one thread) are set to $\{64, 48, 32, 24, 16, 8\}$cm, where $v_{max} = 64$cm and $v_{min} = 8$cm. For ours (two threads), we replace the last 8cm volume with 16cm volume. In all our experiments, we fix the iteration number of initialization $N_i$ to 1500, the number of sampling pixels $|\mathcal{M}|$ to 3000, the size of the window $|\mathcal{W}|$ to 21. The iteration number of window optimization for ours(one thread) $N_w$ is 100. In our two-thread version, we change $N_w$ to 20 for tracking. We follow NICE-SLAM and perform mapping once every 5 frames. Please refer to the Appendix A.5 for more details. We use the Adam (Kingma & Ba, 2014) optimizer to optimize both the implicit map representation and the camera poses, with learning rates of 0.001 and 0.01, respectively. The $\alpha_{\text{warping}}, \alpha_{\text{render}}, \alpha_{\text{smooth}}$ is 0.5, 0.1, 0.01, respectively. We apply a post-optimization at the end of the sequence to improve the quality of the mesh. Note a similar optimization is included

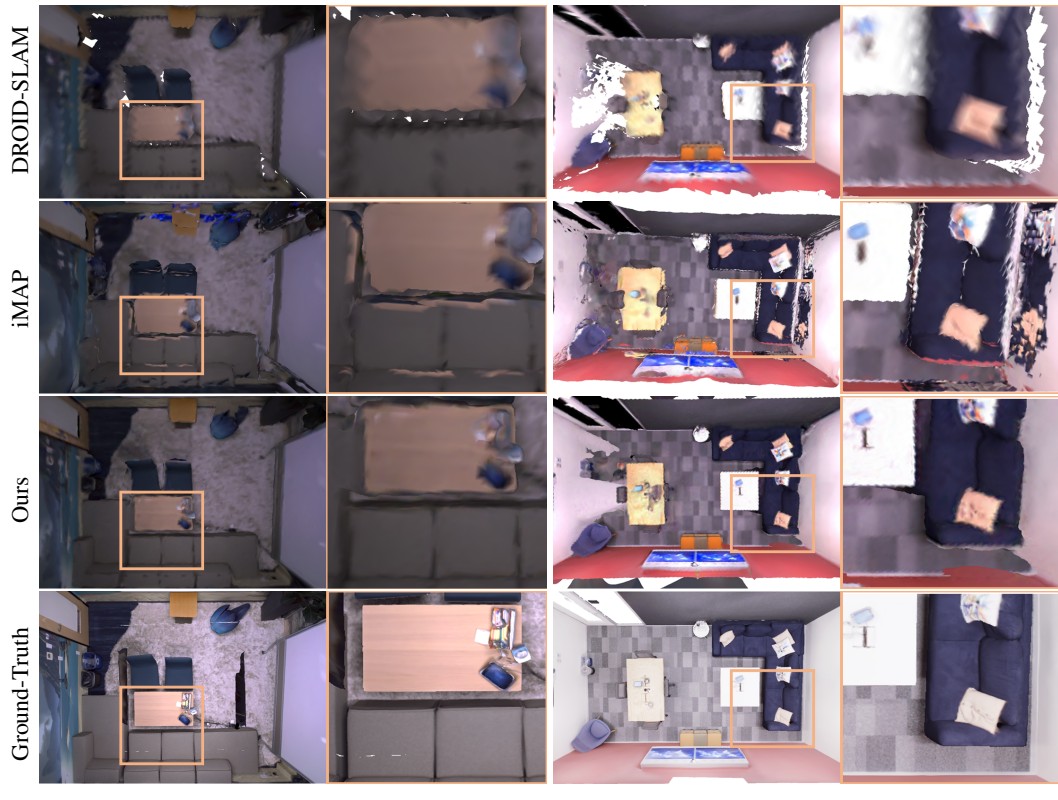

office0                                         office2

Figure 2: Visual comparison of the reconstructed meshes on the *Replica* datasets. Our results are more complete and have sharper textures, which indicate more precise surface shapes. More examples are in the appendix.

| Method | fr1/desk | fr2/xyz | fr3/office | Mean |
|---|---|---|---|---|
| iMAP (Sucar et al., 2021) | 4.9 | 2.0 | 5.8 | 4.2 |
| iMAP*(Sucar et al., 2021) | 7.2 | 2.1 | 9.0 | 6.1 |
| DI-Fusion(Huang et al., 2021) | 4.4 | 2.3 | 15.6 | 7.4 |
| NICE-SLAM(Zhu et al., 2022) | 2.7 | 1.8 | 3.0 | 2.5 |
| BAD-SLAM(Schöps et al., 2019) | 1.7 | 1.1 | 1.7 | 1.5 |
| Kintinuous(Whelan et al., 2012) | 3.7 | 2.9 | 3.0 | 3.2 |
| ORB-SLAM2(RGBD) | **1.6** | **0.4** | **1.0** | **1.0** |
| ORB-SLAM2(RGB)(Mur-Artal & Tardós, 2017) | 1.9 | 0.6 | 2.4 | **1.6** |
| DROID-SLAM(Teed & Deng, 2021) | **1.8** | **0.5** | 2.8 | 1.7 |
| Ours (one thread) | 2.0 | 0.6 | **2.3** | **1.6** |

Table 2: Camera tracking results on the *TUM-RGBD* dataset. The upper part shows RGB-D SLAM methods, while the lower part shows RGB SLAM methods. "∗" denotes the version reproduced by NICE-SLAM. The best results are **bolded**. The metric unit is [cm].

in iMAP (Sucar et al., 2021) and NICE-SLAM (Zhu et al., 2022). All metrics reported in this paper are an average of the results for 5 runs.

## 4.1 EVALUATION OF MAPPING AND TRACKING

**Mapping** To compare the quality of the reconstructed scene maps, we first evaluate the RGB-D methods on the *Replica* dataset, including NICE-SLAM (Zhu et al., 2022), iMAP (Sucar et al., 2021), and DI-Fusion (Huang et al., 2021). Note all these methods use additional depth image inputs. We further compare with the RGB-only method, DROID-SLAM (Teed & Deng, 2021), where the TSDF-Fusion (Curless & Levoy, 1996) is employed to compute a mesh model of the scene map with a voxel grid resolution of $256^3$ using the estimated depth maps.

We use three metrics to evaluate the recovered scene map on *Replica* with $200,000$ sampled points from both the ground-truth and the reconstructed mesh. (i) *Accuracy* (cm), the mean Euclidean

| Method | V101 | V102 | V103 | V201 | V202 | V203 |
|---|---|---|---|---|---|---|
| TartanVO(Wang et al., 2020) | 0.389 | 0.622 | 0.433 | 0.749 | 1.152 | 0.680 |
| SVO(Forster et al., 2014) | 0.070 | 0.210 | X | 0.110 | 0.110 | 1.080 |
| DSO(Engel et al., 2017) | 0.089 | 0.107 | 0.903 | **0.044** | 0.132 | 1.152 |
| DROID-VO(Teed & Deng, 2021) | 0.103 | 0.165 | 0.158 | 0.102 | 0.115 | 0.204 |
| DPVO(Teed et al., 2022) | **0.053** | 0.158 | 0.095 | 0.095 | **0.063** | 0.310 |
| Ours (one thread) | 0.068 | **0.079** | X | 0.053 | 0.178 | X |

Table 3: Camera Tracking Results on EuRoC. The best results are **bolded**. The failure cases are marked as X. The metric unit is [m].

distance between sampled points from the reconstructed mesh to their nearest points on the GT mesh. (ii) *Completion* (cm), the mean Euclidean distance between sampled points from the GT mesh to their nearest points on the reconstructed mesh. (iii) *Completion Ratio* (%), the percentage of points in the reconstructed mesh whose nearest points on the GT mesh is within 5 cm. The unseen region is removed if it is not inside any camera viewing frustum.

As shown in the lower part of Table 1, our method outperforms iMAP and DROID-SLAM on mapping. Although our results are slightly inferior to those of NICE-SLAM and DI-Fusion, this is reasonable since they have additional depth images as input. A visual comparison with iMAP and DROID-SLAM is provided in Figure 2. From the zoomed regions, we can tell that our method generates more complete and accurate models than iMAP and DROID-SLAM. Additional visual comparisons are provided in the appendix.

**Tracking** To evaluate the quality of camera tracking, we compare our method with many recent visual SLAM methods with RGB or RGB-D inputs. For a fair comparison, we use the official implementation of DROID-SLAM to produce its result and cite other results from NICE-SLAM and DPVO (Teed et al., 2022). As for the evaluation metric, we adopt the RMSE of Absolute Trajectory Error (ATE RMSE) (Sturm et al., 2012b). The estimated trajectory is aligned with the ground truth with scale correction before evaluation.

The upper part of Table 1 shows the tracking results on the *Replica* dataset. Our method clearly outperforms NICE-SLAM, though it has additional depth input. Our tracking performance is similar to DROID-SLAM on this dataset.

Table 2 shows our results on the *TMU RGB-D* dataset. We report results on the same three examples as iMAP. As shown in the table, our method outperforms all previous learning-based methods, including iMAP (Sucar et al., 2021), NICE-SLAM (Zhu et al., 2022), DI-Fusion (Huang et al., 2021), and has comparable performance with robust conventional methods like BAD-SLAM (Schöps et al., 2019), Kintinuous (Whelan et al., 2012), and ORB-SLAM2 (Mur-Artal & Tardós, 2017). Furthermore, our method does not include global bundle adjustment or loop closure, which is common in the conventional SLAM system. Our method achieves reasonable tracking performance while enjoying the benefits of neural implicit map representation, which can produce a watertight scene model or predict unobserved regions.

Table 3 shows experiment results on the *EuRoC* dataset. We compare the tracking results with several strong baselines including TartanVO (Wang et al., 2020), SVO (Forster et al., 2014), DSO (Engel et al., 2017), DROID-VO (Teed & Deng, 2021), and DPVO (Teed et al., 2022). Our method generates comparable results in successful cases. Note there are two failure cases, where our method loses camera tracking due to the fast camera motion which leads to small view overlap between neighboring frames.

## 4.2 ABLATION STUDY

To inspect the effectiveness of the components in our framework, we perform the ablation studies on the `office-0` sequence of the *Replica* dataset.

**Hierarchical Feature Volume** To evaluate the effectiveness of our hierarchical feature volume, we experiment with different levels of hierarchies. We vary the number of hierarchies $L$ and the number of feature channels $C$ to keep the total number of learnable parameters similar. Table 4 shows the results of different settings. Clearly, the system performs worse when there are less hierarchies of feature volumes, even though a longer feature is learned to compromise it. Note the system fails

| | Configurations | | | | $(L, C)$ | | | | | w/o MLP |
|---|---|---|---|---|---|---|---|---|---|---|---|
| | | (6,1) | (4,4) | (2,8) | (1,16) | (6,2) | (6,4) | (6,8) | (6,16) | |
| ATE RMSE(cm)↓ | MEAN | 0.67 | 0.76 | 2.27 | X | 0.63 | 0.69 | 0.61 | 0.66 | 8.92 |
| | STD | 0.14 | 0.18 | 0.25 | X | 0.12 | 0.08 | 0.11 | 0.09 | 1.49 |
| Accuacry(cm)↓ | MEAN | 2.60 | 3.39 | 6.78 | X | 2.86 | 2.35 | 2.96 | 2.48 | 11.43 |
| | STD | 0.56 | 0.91 | 0.95 | X | 0.49 | 0.33 | 0.40 | 0.12 | 3.17 |

Table 4: Ablation study of different configurations of the feature volume. The parameters $L$ and $C$ are the numbers of hierarchies and the number of the feature channel, respectively. We test on `office-0` for 5 times and calculate the mean and standard deviation of the camera tracking error and map accuracy.

| Method | Memory(MB)↓ | FLOPs($\times 10^3$)↓ | Mapping(ms)↓ | Tracking(ms)↓ |
|---|---|---|---|---|
| iMAP | 1.02 | 443.91 | $448(1000, 10)$ | $101(200, 6)$ |
| NICE-SLAM | 12.0(16cm) | 104.16 | $130(1000, 10)$ | $47(200, 6)$ |
| Ours (two threads) | 9.1(16cm) | 18.76 | $330(3000, 100)$ | $72(3000, 20)$ |
| Ours (one thread) | 29.1(8cm) | 18.76 | $335(3000, 100)$ | |

Table 5: Memory, computation, and running time. We report the runtime in `time(pixel, iter)` format for a fair comparison. Please refer to the appendix for more information about our two-threads version.

to track camera motion at textureless regions when $L$ is set to 1. This experiment demonstrates the effectiveness of fusing features from different scales to better capture scene geometry.

**Number of Feature Channel** We further vary the number of feature channel $C$ with $L$ fixed at 6. Table 4 shows similar system performance is achieved for $C = 2, 4, 8, 16$. Therefore, to keep our method simple, we select $C = 1$ in experiments to reduce computation cost.

**MLP Decoder** To study the impact of the MLP decoder, we also experiment without the MLP decoder, where the 1-channel feature can be regarded as the occupancy directly. At a 3D point, we sum the occupancies from all hierarchies for the final result. The color is computed similarly with another 6-level volume of 3 channels. As shown in the right of Table 4, the system performance drops significantly without the MLP decoder.

### 4.3 MEMORY AND COMPUTATION ANALYSIS

Table 5 analyzes the memory and computation expenses. The memory cost is evaluated on the `office-0` sequence of the *Replica* dataset. We use the same bounding box size of the feature volume in both NICE-SLAM and our method. iMAP is the most memory efficient with only 1.02MB map size. However, as discussed in NICE-SLAM, it cannot deal with larger scenes. Our one thread version, i.e. ours (one thread), recovers the scene map at higher resolution of 8cm and requires a larger map of 29.1 MB, while NICE-SLAM works at 16cm map resolution and takes only a 12 MB map. Our two thread version, i.e. ours (two threads), uses the same scene map resolution of 16cm, and is more memory efficient with map size of 9.1 MB.

Table 5 also reports the number of FLOPs for computing the color and occupancy at a single 3D point. Our method requires much less FLOPS, benefiting from our short features and single shallow MLP decoder. Table 5 further reports the computation time of processing one input frame. We report the results in `time(pixel, iter)` format for comparison. Without depth input, our method samples more pixels and takes more optimization iterations, while our method is still faster than iMAP. Our running time is reported with one or two RTX 2080TI GPUs, while the running time of NICE-SLAM and iMAP are cited from their papers with an RTX 3090 GPU.

## 5 CONCLUSION

This paper introduces a dense visual SLAM method working with regular RGB cameras with neural implicit map representation. The scene map is encoded with a hierarchical feature volume together with an MLP decoder, which are both optimized with the camera poses simultaneously through back-propagation. To deal with regular RGB cameras, a sophisticated warping loss is designed to enforce multi-view geometry consistency during the optimization. The proposed method compares favorably with the widely used benchmark datasets, and even surpasses some recent methods with RGB-D inputs.

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

# A    APPENDIX

In the appendix, we present the following:

## A.1    FEATURE VOLUME INITIALIZATION

We initialize the parameters of all feature volumes under a normal distribution, while the mean and the standard deviation of the distribution are $0$ and $0.001$, respectively.

## A.2    POST OPTIMIZATION IN OURS(ONE THREAD)

To enhance the geometry and color consistency throughout the entire scene, we also introduce a post-optimization module similar to the one in NICE-SLAM (Zhu et al., 2022) and iMAP (Sucar et al., 2021). This module is optional at the end of processing an input video clip to improve the final mesh quality without heavy computation. This post-optimization module optimizes the implicit scene representation with the camera poses fixed using all keyframes in an alternative way. In each iteration, we sample a random keyframe $k$ and select 20 keyframes from all other keyframes with at least $70\%$ overlap with frame $k$. We optimize the implicit function using all keyframes $1,000$ iterations.

## A.3    ABLATION ON WARPING LOSS

To see the impact of utilizing multi-scale patch warping photometric loss, we run an ablation study on different settings of the warping loss. We evaluate different configurations during the initialization stage. We mark our default configuration as the baseline: using a multi-scale patch warping loss defined as Equation 7 where $\mathcal{S} = \{1, 5, 11\}$. We then compare it with two different patch size configurations, where $\mathcal{S} = \{1\}$ and $\mathcal{S} = \{1, 11\}$, respectively. Note that the setting of $\mathcal{S} = \{1\}$ degrades to the pixel-wise warping loss as Equation 5. As shown in Table 6, our default configuration achieves 1.85cm in terms of the average depth error. The setting of $\mathcal{S} = \{1, 11\}$ is slightly inferior to the baseline, while the setting of pixel-wise warping fails to recover an accurate depth. This is also illustrated in the qualitative comparison in Figure 3.

## A.4    MORE ABLATION ON FEATURE CHANNEL

We further verify the number of feature channels with the initialization stage. As shown in the right of Table 6, increasing the number of features $C$ does not improve the quality of the initialized depth. Note that while the setting $(L = 1, c = 16)$ can still initialize the reconstruction, the camera tracking soon fails in textureless regions. This ablation further justifies our choice of $C = 1$.

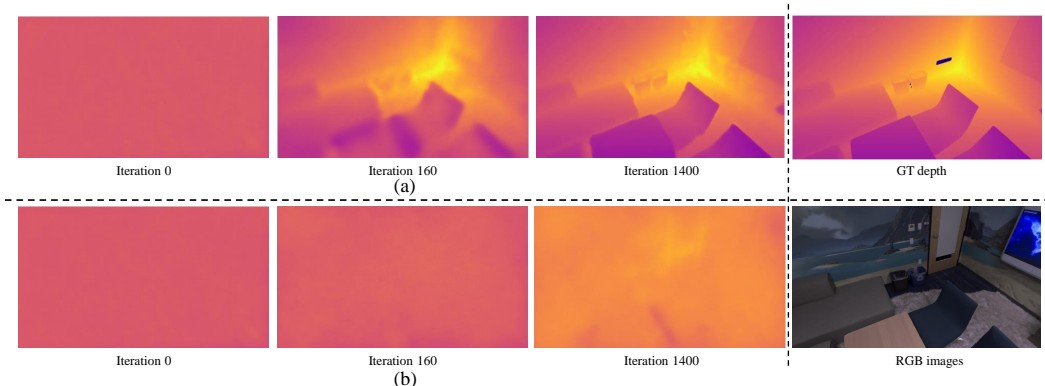

Figure 3: The visualization of the initialized depth map during optimization iterations. (a) The default configuration with multiple patch-wise warping loss, $\mathcal{S} = \{1, 5, 11\}$. (b) The pixel-wise warping loss, $\mathcal{S} = \{1\}$.

| Depth Error(cm)↓ | $\mathcal{S}$ | | | $(L, C)$ | | | |
|---|---|---|---|---|---|---|---|
| | $\{1, 5, 11\}^*$ | $\{1, 11\}$ | $\{1\}$ | $(6, 1)^*$ | $(6, 4)$ | $(6, 16)$ | $(1, 16)$ |
| MEAN | 1.85 | 3.73 | 35 | 1.85 | 1.72 | 1.93 | 1.83 |
| STD | 0.15 | 0.85 | 3.65 | 0.15 | 0.09 | 0.16 | 0.19 |

Table 6: Mean depth errors of initialized map for different settings in the ablation study. The configuration of our method is marked with $^*$. We report the mean and std of depth error on 5 runs of `office-0` on *Replica*.

| Method | *Replica* | | *TUM-RGBD* | |
|---|---|---|---|---|
| | Mapping | Tracking | Mapping | Tracking |
| NICE-SLAM | (1000,60) | (200,10) | (5000,60) | (5000,200) |
| Ours(two threads) | (3000,100) | (3000,20) | - | |
| Ours(one thread) | | (3000,100) | | |

Table 7: (Pixel, Iteration) used in different datasets. NICE-SLAM needs more pixels and iterations on *TUM-RGBD*, while ours(one thread) keeps the same configuration on all datasets.

## A.5 TWO THREADS FOR TRACKING AND MAPPING, OURS(TWO THREADS)

We provide more details about our configuration which runs tracking and mapping on two threads. The thread of tracking is similar to ours(one thread), while we only optimize the camera poses:

$$\arg\min_{\mathcal{P}_k, k \in \mathcal{W}} \alpha_{\text{warping}} L_{\text{warping}} + \alpha_{\text{render}} L_{\text{render}}, \tag{10}$$

where $\mathcal{W}$ is the window of the frames defined in Section 3.3, and we sample high gradient pixels following the strategy in iNeRF (Yen-Chen et al., 2021). We sample $3,000$ pixels and optimize 20 iterations for each frame.

In the mapping thread, we optimize both cameras poses and the parameters of the scene simultaneously:

$$\arg\min_{\Theta, \mathcal{P}_k, k \in \mathcal{W}_{\text{mapping}}} \alpha_{\text{warping}} L_{\text{warping}} + \alpha_{\text{render}} L_{\text{render}} + \alpha_{\text{smooth}} L_{\text{smooth}}, \tag{11}$$

where $\mathcal{W}_{\text{mapping}}$ is the keyframe window mentioned in A.2, and we random sample $3,000$ pixels for mapping.

The major difference between the post-optimization in A.2 and the mapping thread is that we only run pots-optimization at the end of the sequence. Furthermore, we do not optimize the camera pose during post-optimization in ours(one thread).

The memory and time cost of both ours(one thread) and ours(two threads) are listed in Table 5. To fairly compare the memory consumption between our method and NICE-SLAM (Zhu et al., 2022), we replace the volume with size 8cm by 16cm. NICE-SLAM (Zhu et al., 2022) and ours have the same resolution in this configuration.

We report our performance of ours(two threads) and ours(one thread) on the Replica Dataset in Table 1. The result of ours(two threads) is slightly inferior compared with ours(one thread), while it still outperforms another RGB method.

In Table 7, we report the number of pixels and iterations used in different datasets. Our method keeps the same setting for all datasets, while NICE-SLAM requires more pixels and iterations on the real-world dataset.

## A.6 MESH EXTRACTION

Our implicit map representation contains a hierarchical feature volume and an MLP decoder. We can compute occupancy for each point in the 3D space from our implicit map representation. Our method can easily predict the occupancy for unseen points. We use the marching cubes algorithm (Lorensen & Cline, 1987) to create a mesh for visualization and evaluation.

## A.7 MORE VISUALIZATION RESULT

To see more shape details on *Replica* (Straub et al., 2019), we also render these extracted meshes at a close viewpoint and compare our results with those from iMAP (Sucar et al., 2021) and DROID-SLAM (Teed & Deng, 2021) in Figure 4. It is clear that our method recovers more shape details and can fill in unobserved regions with the neural implicit map.

We provides visualization of all reconstructed scenes in *Replica* (Straub et al., 2019) in Figure 5 and 6.

In Figure 7, we show two mesh reconstructions on the *TUM RGB-D* dataset. Our method could recover high-quality mesh due to accurate camera poses and the smaller voxel size.

In Figure 8, we run our method on `pikachu_blue_dress1_camera3` in (Shrestha et al., 2022) with different voxel size. We show the results when the smallest volume size is $8, 2, 0.5$cm, respectively. Our method could recover fine detail of the small object if we increase the resolution of volume.

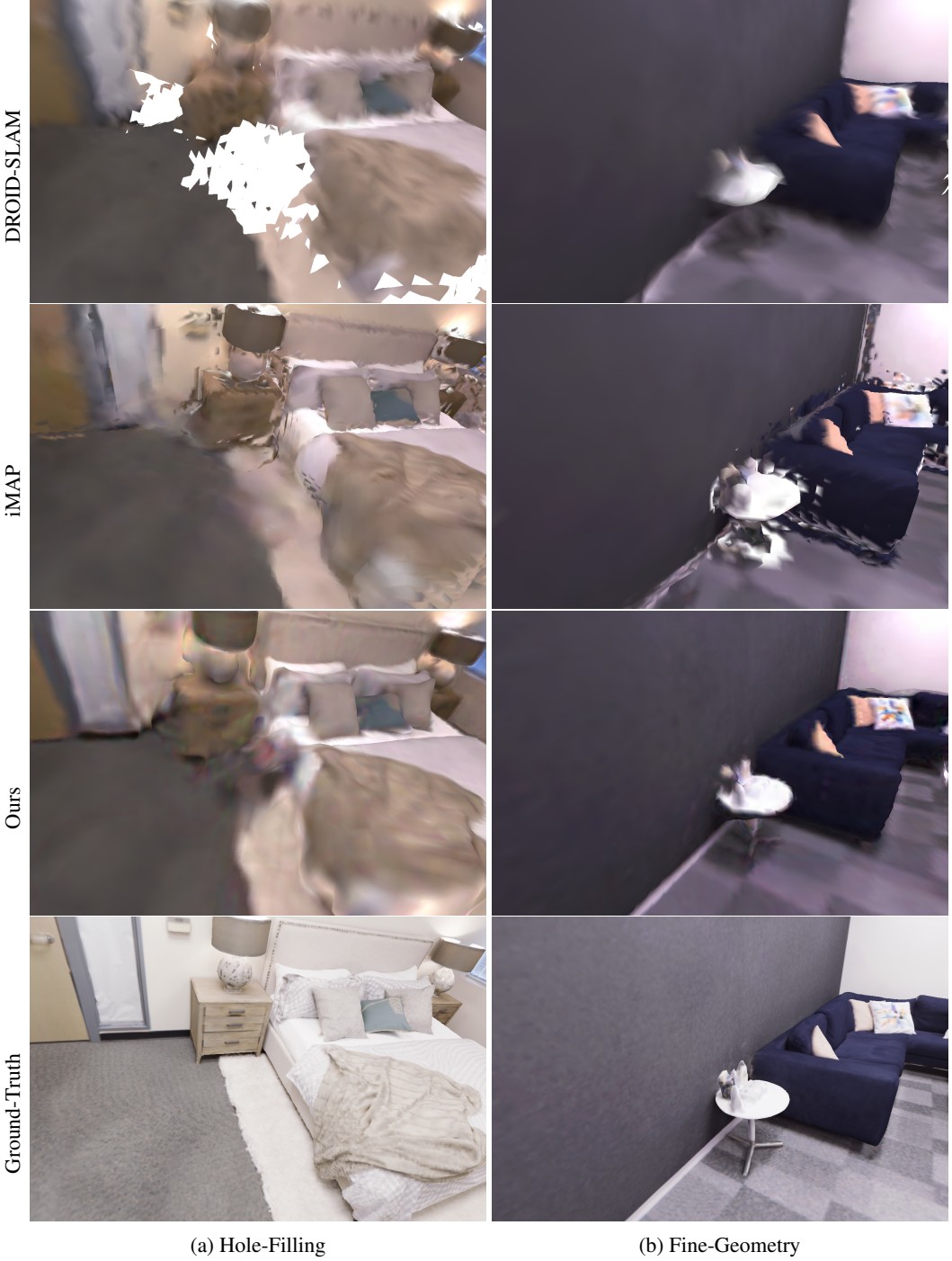

(a) Hole-Filling                  (b) Fine-Geometry

Figure 4: Rendering of the reconstructed mesh from the close viewpoint. Our results are more complete and accurate than those from iMAP and DROID-SLAM.

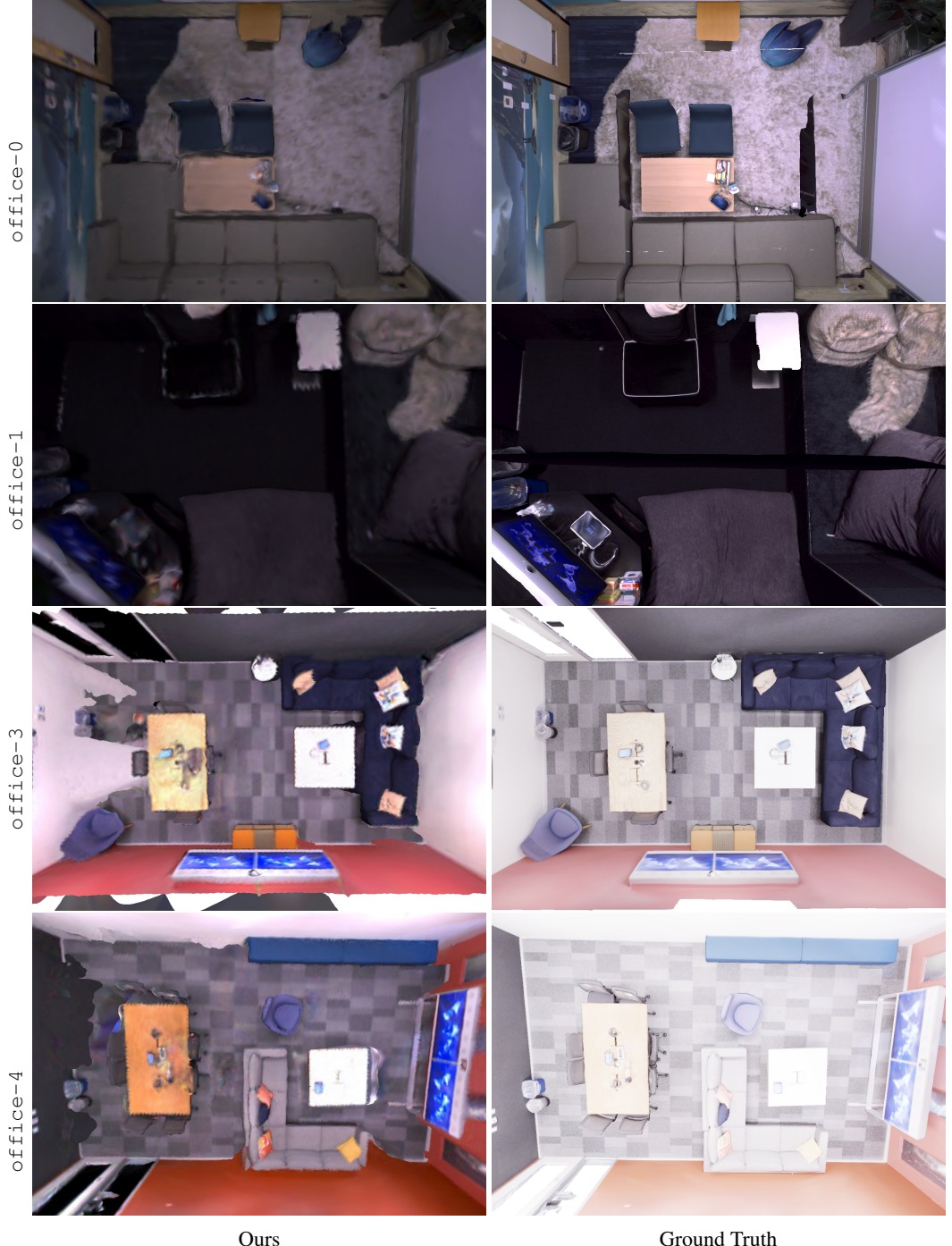

Figure 5: Visual comparison of the reconstructed meshes on the *Replica* datasets.

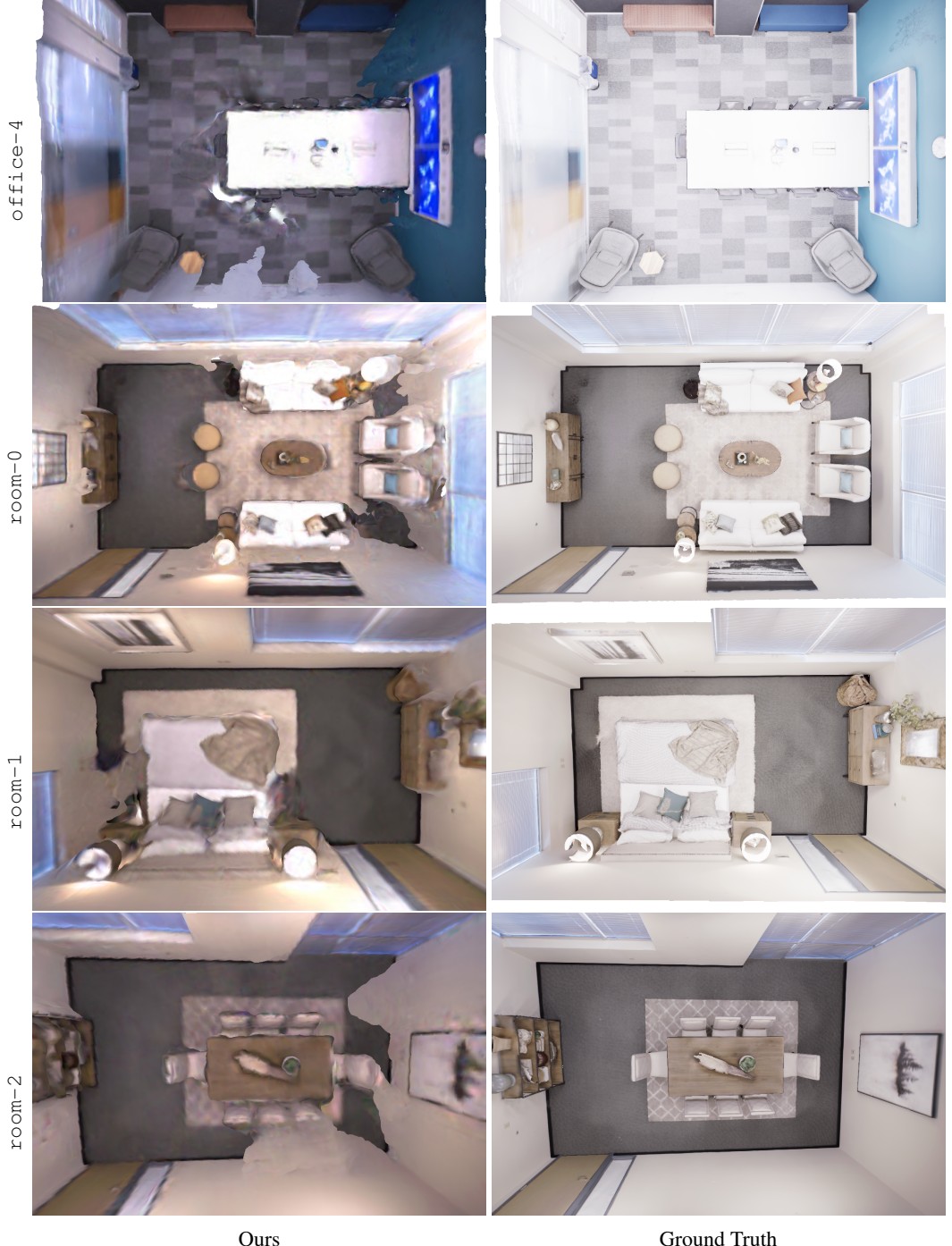

Ours                    Ground Truth

Figure 6: Visual comparison of the reconstructed meshes on the *Replica* datasets.

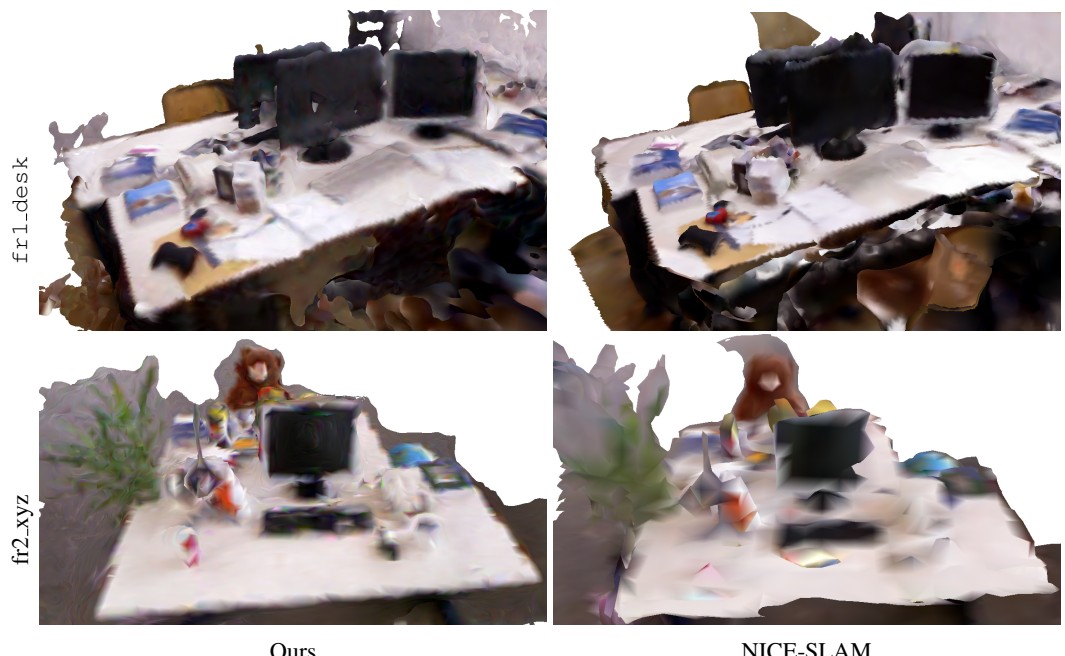

Ours                                                            NICE-SLAM

Figure 7: The reconstructed mesh on *TUM-RGBD* dataset.

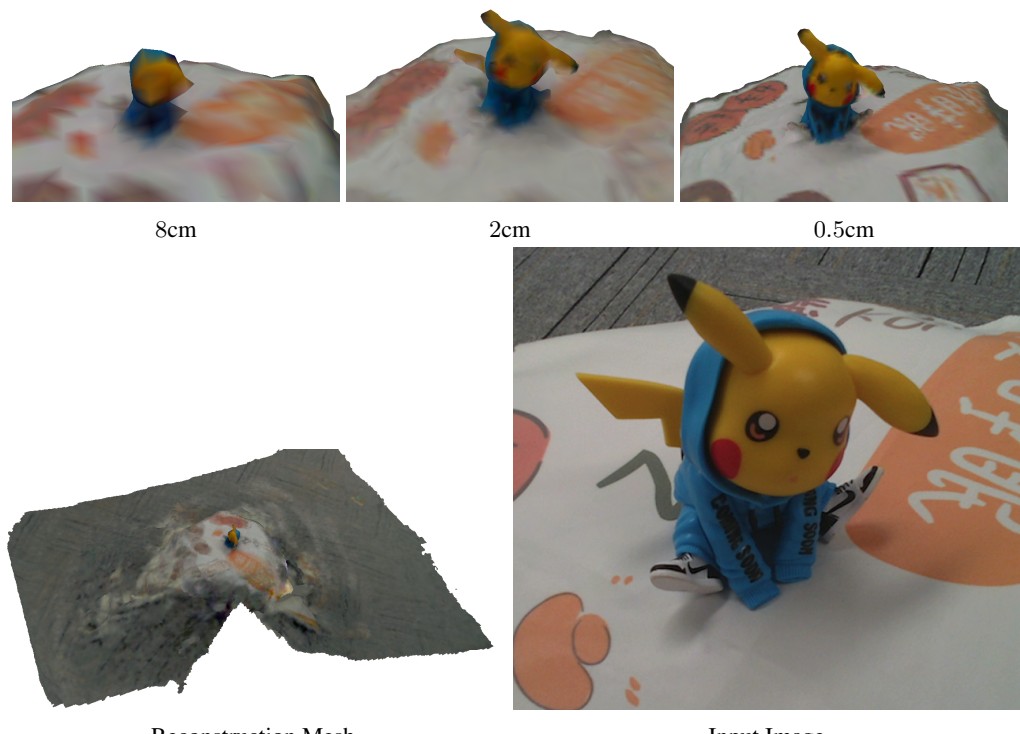

8cm                             2cm                             0.5cm

Reconstruction Mesh                              Input Image

Figure 8: The reconstructed mesh of `pikachu_blue_dress1_camera3` on *MeshMVS* (Shrestha et al., 2022) dataset using different size of voxel.

