# OpenReview forum: "DENSE RGB SLAM WITH NEURAL IMPLICIT MAPS"
_ICLR.cc/2023/Conference — ICLR 2023 poster_

### Official Review · Reviewer_K6me · 2022-10-22

**Confidence:** 3
**Clarity, Quality, Novelty And Reproducibility:** The calrity, quality, novelty and rep…
**Correctness:** 3
**Technical Novelty And Significance:** 3
**Empirical Novelty And Significance:** 2
**Recommendation:** 6

**Strength And Weaknesses:**

strength:

(1)	The main contribution of the paper is the proposed dense SLAM system with implicit map representation, which allows SLAM system with implicit map representation to take only RGB images as input.

(2)	Experiments demonstrate that the proposed method gives competitive pose and depth accuracy to previous both classic and learning-based methods.

Weakness:

(1)	Higher memory and computational cost. The proposed method adopts hierarchical feature volumes with more levels to make the system work, which inevitably brings additional memory and time cost.

(2)	Missing explanations of some details.

a.	One thread for joint pose and map optimization. In many classic VO/SLAM systems (e.g., ORB-SLAM) and learning-based models (e.g., NICE-SLAM), two or more threads are used for tracking and mapping respectively to speed up. Is it possible to use a thread for tracking by taking only important pixels into consideration and another slower thread for mapping in this system?

b.	In Eq (4), the paper uses randomly sampled M pixels for photometric rendering loss computation. In fact, because of the noise or the dynamic objects, not all pixels can provide the equal contribution to the constraint, so it might be better to use more stable pixels as in Eq (7).

(3)	Discussion of related works of NeRFs without pose as input.
This work is also related to works of NeRFs which do not require poses as input, such as R1.

R1: Barf: Bundle-adjusting neural radiance fields, Lin et al., ICCV 2021.

**Summary Of The Paper:**

In this paper, the author proposes a RGB SLAM model with neural implicit map representation. In contrast to previous competitors, e.g., iMap and NICE-SLAM requiring depth maps as input, the proposed approach takes only RGB image sequences as input and leverages the hierarchical feature volumes with more levels to boost the performance. Experiments show that the proposed method achieves more accurate poses than NICE-SLAM and better depth maps than DROID-SLAM, but at the cost of higher memory and running time.

**Summary Of The Review:**

As this paper is the first dense RGB SLAM with neural implicit map representation, I tend to accept this paper.

---

> ### Author Response · Authors · 2022-11-18
> **Response to reviewer K6me**
>
> We thank the reviewer for the valuable suggestions and address the reviewer's concerns as follows.
>
> **W1 Memory and runtime cost**
>
> As discussed in Section 4.3, our method requires 29.1 MB and 9.1 MB maps to achieve $8$cm and $16$cm resolutions, respectively. NICE-SLAM requires a 12.0 MB map to achieve the $16$cm resolution. Our method is in fact more memory efficient than NICE-SLAM under the same map resolution. Our method is indeed slower than NICE-SLAM, mainly because we are dealing with a more challenging problem. As shown in Table 5, we need many more samples and iterations to solve the tracking and mapping problem.
>
> **W2.a Separate threads for tracking and mapping**
>
> Thanks for the suggestion! In the revised version, we provide a two-thread implementation, i.e. ours (two threads), with implementation details explained in appendix A.5. The memory and the time cost of ours (two threads) are shown in Table 5, which is more efficient than the one-thread version. Its corresponding results are listed in Table 1.
>
> In this two-thread version, we follow iNeRF to sample pixels with higher gradients for tracking to speed up the convergence. More details are explained in appendix A.5. Our mapping still uses uniform random sampling to capture textureless regions.
>
> **W2.b Stable pixel for optimization**
>
> Choosing only stable pixels on static objects is a good idea. NICE-SLAM uses the input depth maps to filter pixels on moving objects. But this filtering is difficult with just RGB input. We leave this topic for future research.
>
> **W3 Missing reference of NeRF without pose**
>
> Thanks for pointing it out. We have cited and discussed `NeRF without pose methods` in the revised related work section.

---

### Official Review · Reviewer_pLyh · 2022-10-23

**Confidence:** 4
**Correctness:** 3
**Technical Novelty And Significance:** 3
**Empirical Novelty And Significance:** 3
**Recommendation:** 8

**Clarity, Quality, Novelty And Reproducibility:**

**Clarity of presentation**

There are multiple statements that need clarification and / or revision:
* The statements "However, RGB-D cameras are limited to indoor scenes and are much less energy efficient for mobile devices." and "However, they both require RGB-D sequences as input, which limits their application in outdoor scenes or mobile devices with a tight power budget." are used to motivate using a RGB-only approach. Yet, these statements ignore that depth sensors can operate outdoors, e.g., when based on time-of-flight measurements such as the depth sensor used by HoloLens 2, and that such sensors are available on mobile devices with tight power budgets, e.g., HoloLens 2. While it is true that RGB cameras are more energy efficient, I don't think this implies that RGB-SLAM approaches are more energy efficient than RGB-D-SLAM methods as the former often require compute to obtain depth measurements. Case in point, the proposed approach is clearly too powerhungry to be applicable on mobile devices. Furthermore, the proposed approach is only evaluated in indoor scenes.
* I don't see how the proposed feature volume is "fundamentally different": 1) The statement "Firstly, the decoders in NICE-SLAM (Zhu et al., 2022) are pretrained, which might cause problems when generalizing to different scenes, while our method learns the scene features and decoders together on the fly." seems self-contradictory: if the proposed approach learns the features and decoders on the fly per scene, then it does not attempt to generalize, so criticizing another method for attempting to generalize seems strange to me. It is furthermore not a difference in the feature volume. 2) The difference between aggregating features and computing occupancy from the aggregated feature instead of computing occupancies first and then aggregating them seems more like a design / implementation choice than a fundamental difference of the underlying feature volume.
* The paper states that "In this way, we save many unnecessary occupancy evaluations and thus can afford to have much more layers of feature hierarchy." While true, the overall run-time and memory requirements of the proposed approach are still higher than NICE-SLAM.
* The paper states that "While NICE-SLAM (Zhu et al., 2022) only optimizes two feature volumes with voxel sizes of 32cm and 16cm, our method solves six feature volumes from 8cm to 64cm. Our fusion of features across many different scales leads to more robust and accurate tracking and mapping as demonstrated in experiments." I don't think the last claim holds as the paper compares a RGB- and a RGB-D-based approach, from which I don't clearly see that the fusion of the features is the main reason that the proposed approach is more accurate and robust (setting aside that this statement is not always true as shown in the experiments).

**Originality of the work**

Please see W4 above for details. Overall, while the individual components might not be novel themselves, I think their combination into a RGB-based system is interesting and the proposed approach and its results will be valuable for the community.

**Strength And Weaknesses:**

**Strength**:

S1) The proposed approach is technically sound and its description in the paper is easy to follow.

S2) Each part of the proposed approach is well-motivated.

S3) The experimental results show that the proposed approach performs well on multiple datasets and is competitive with RGB-D-based methods.

**Weaknesses**:

W1) There are multiple statements that need to be clarified and / or revised (see below).

W2) Eq. 5 is wrong. It should be $q_{k \rightarrow l} = K_l \tilde{R}_l^T \left(\tilde{R}_k K_k^{-1} q_k^\text{homo} \tilde{D}_q + \tilde{t}_k - \tilde{t}_l \right)$.

W3) Some design choices are not fully motivated: For the photometric warping loss, why not use the fact that the neural implicit representation also provides normal information to do a perspective warp of the patch instead of using a fronto-parallel warp? Why use SSIM and not any other robust loss (e.g., see [Park et al., Illumination change robustness in direct visual SLAM, ICRA 2017])?

W4) Many components described in Sec. 3.1 are based on or inspired by prior work but are missing references. E.g., there is prior work on using multi-resolution volumes in combination with a MLP, e.g., NICE-SLAM, and work on explicitly storing the required information in the voxel volume (thus foregoing the use of a MLP), e.g., [Fridovich-Keil et al., Plenoxels Radiance Fields without Neural Networks, CVPR 2022] It would be good to acknowledge this prior work here. Similarly, using a photometric warping loss is common when training neural networks for monocular depth prediction, defining the photometric warping loss over patches is standard in the multi-view stereo literature, and depth map regulation is a standard loss term in networks that predict monocular depth. I don't think it is an issue that the paper builds upon existing work, but existing ideas should be clearly acknowledged.

W5) It is unclear to me how the weights for Eq. 9 are chosen and how sensitive the proposed approach is to their choice.

W6) Given that [Schoeps et al., CVPR 2019] showed that RGB-D-based methods perform worse on TUM than RGB-based methods, attributed due to the rolling shutter sensors used to capture the datasets, why not chose a dataset that enables a better comparison, e.g., the one from [Schoeps et al., CVPR 2019]? Why not compare against ORB-SLAM2 (or better ORB-SLAM3) in RGB- instead of RGB-D-mode to compare against a state-of-the-art feature-based approach?

W7) The paper states that "Our method is implemented with PyTorch (Paszke et al., 2017) and runs on a server with two NVIDIA 2080Ti GPUs." and that "Our running time is reported with an RTX 2080TI GPU". It is unclear to me whether one or two GPUs are used when reporting the running times in Tab. 4. This is important information required to understand the results.

**Summary Of The Paper:**

The paper proposes a SLAM system based on neural implicit representation as the underlying 3D map representation. Inspired by NICE-SLAM, the proposed approach uses a multi-resolution feature grid to represent the scene and jointly updates the camera poses and the map. The main difference to prior work is that the proposed approach operates on RGB instead of RGB-D images. This is achieved by integrating a multi-view stereo loss to enforce consistent depth and color predictions between close-by images. The proposed approach is evaluated on both synthetic and real data and the paper provides an ablation study.

**Summary Of The Review:**

Overall, this is a solid paper that I think is of interest to the community. Most of the concerns raised above can be addressed quite easily. I am happy to increase my ranking if they are addressed. I did not give a higher ranking as I feel that the concerns have to be addressed first.

------- After discussion -----------

During the discussion phase, the authors successfully addressed my concerns. I thus increase my score and recommend to accept the paper.

---

> ### Author Response · Authors · 2022-11-18
> **Response to reviewer pLyh (1/2)**
>
> We thank the reviewer for the valuable suggestions and address the reviewer's concerns as follows.
>
> **W2 Wrong equation**
>
> Thanks for pointing out the mistake in our equation. We have fixed it in the revised version.
>
> **W3 Design motivation**
>
> We agree with the reviewer that a perspective warp is better than a fronto-parallel warp. However, it requires an additional autograd function to compute the surface normal by backpropagation. Backpropagation is very time-consuming (compared to other parts, e.g., network forwarding and parameter updating). Thus, we choose to use a fronto-parallel warp, which still achieves good performance.
>
> We have experimented with a few robust functions, e.g. GradM and ZNCC[4], eventually, we choose SSIM because it can recover a better result with fewer iterations. SSIM performs well as a loss term in many deep learning tasks, e.g., optical flow[2] and depth estimation[3]. Its numerical gradient can be easily computed by the automatic differentiation engine in modern deep learning frameworks. We have discussed this choice in the revised version.
>
>
> **W4 Additional citations**
>
> Thanks for pointing it out. We have added citations and discussions of those works, e.g. NICE-SLAM, Plenoxel, warping-based depth estimation, etc., in the revised version in both the introduction and related work sections.
>
> **W5 Hyperparameter**
>
> We test different combinations of $\alpha_{\text{render}}$, $\alpha_{\text{warping}}$, and  $\alpha_{\text{smooth}}$. Our method is robust to these hyperparameters. Generally, the warping loss is more important to recover a high-quality 3D map. Thus, we set $\alpha_{\text{warping}}$ larger than $\alpha_{\text{render}}$. The regularization term $\alpha_{\text{smooth}}$ is usually small. In all our experiments, they are fixed at $ 0.5, 0.1, 0.01$.
>
> **W6 Dataset and baseline**
>
> To make the comparison with iMAP and NICE-SLAM easy, we follow them to test on the TUM RGB-D and Replica datasets. We have added the results of ORB-SLAM2 in RGB mode on the TUM RGB-D dataset in the revised paper. Again, our method achieves similar tracking performance as it (and also DROID-SLAM).
>
> **W7 the number of GPU**
>
> We have clarified it in the revised version. For our single-thread setting, marked as ours (one thread), we only use one 2080Ti GPU for both mapping and tracking. For the setting of ours (two threads), we run two threads on two 2080Ti GPUs for tracking and mapping, respectively.

---

> ### Author Response · Authors · 2022-11-18
> **Response to reviewer pLyh (2/2)**
>
> **P1 Motivation of RGB SLAM over RGB-D SLAM**
>
> While modern time-of-flight cameras can work to some extent in outdoor environments, their performance under strong ambient illumination is significantly deteriorated. Furthermore, RGB-D cameras often have limited image resolutions for other visual tasks such as object detection/recognition. We have revised the introduction to motivate our RGB-based SLAM.
>
> **P2 Statements of the feature volume**
>
> We agree with the reviewer and have changed the `fundamentally different` to `more suitable to visual SLAM`.
>
> For the decoder, as we discussed in [Review bVvs W1](https://openreview.net/forum?id=QUK1ExlbbA&noteId=Gn8Skh0o-4), the pretrained decoder in NICE-SLAM has difficulties to generalize to other scenes, which is also observed in other works [1]. In comparison, our method learns a decoder on the fly for each specific scene and is free from this problem.
>
> Furthermore, NICE-SLAM needs to evaluate the decoder multiple times to compute the occupancy value. In comparison, our method only evaluates it once and is more computationally efficient.
>
> Last but not least, the number of feature channels is $1$ in our volume, making our method much more efficient in memory and computation. Therefore, our method can afford to have many more sample points and optimization iterations (see Table 5), which are important to make our algorithm work without depth input.
>
>
> **P3 Memory and runtime**
>
> Our method is slower than NICE-SLAM because we are solving a more challenging problem, i.e. RGB SLAM vs RGB-D SLAM. As discussed in [Reviewer W3wZ W2](https://openreview.net/forum?id=QUK1ExlbbA&noteId=La34MUiJpEt), the additional depth input can accelerate optimization. Compared with NICE-SLAM, our formulation needs only 1/6 FLOPS to evaluate the occupancy and color at a sampled point. But as shown in Table 5, our problem requires many more sample points and iterations since there is no depth input.
>
> **P4 Fusion of the Feature**
>
> In Table 4 in the main paper, we experimented with different numbers of feature hierarchies. The mean RMSEs measuring camera tracking accuracies with 6, 4, and 2 layers of features are 0.67cm, 0.76cm, and 2.27cm respectively. Similarly, the mapping errors are 2.60cm, 3.39cm, and 6.78cm. We believe this is strong evidence that more feature hierarchies can boost SLAM performance. This is also intuitively reasonable since features at different hierarchies encode shape information at different scales, this multi-scale feature can better constrain the pose and map optimization.
>
> [1] Yang, X., Li, H., Zhai, H., Ming, Y., Liu, Y., & Zhang, G. (2022). Vox-Fusion: Dense Tracking and Mapping with Voxel-based Neural Implicit Representation. arXiv preprint arXiv:2210.15858.
>
> [2] Luo, K., Wang, C., Liu, S., Fan, H., Wang, J., & Sun, J. (2021). Upflow: Upsampling pyramid for unsupervised optical flow learning. In Proceedings of the IEEE/CVF Conference on Computer Vision and Pattern Recognition (pp. 1045-1054).
>
> [3] Godard, C., Mac Aodha, O., Firman, M., & Brostow, G. J. (2019). Digging into self-supervised monocular depth estimation. In Proceedings of the IEEE/CVF International Conference on Computer Vision (pp. 3828-3838).
>
> [4] Park, S., Schöps, T., & Pollefeys, M. (2017, May). Illumination change robustness in direct visual slam. In 2017 IEEE international conference on robotics and automation (ICRA) (pp. 4523-4530). IEEE.

---

### Official Review · Reviewer_W3wZ · 2022-10-26

**Confidence:** 3
**Correctness:** 3
**Technical Novelty And Significance:** 3
**Empirical Novelty And Significance:** 3
**Recommendation:** 6

**Clarity, Quality, Novelty And Reproducibility:**

Clarity, Quality, Novelty:
The overall paper is well written with adequate experiments. Details could be found in last section.

Reproducibility: The codes are promised to be released so there should be possible to reproduce the work.


**Strength And Weaknesses:**

Strengths:

Monocular dense mapping of indoor scenes is an interesting and important research direction. This paper builds upon previous work on RGB-D neural mapping and extends them to monocular setup with proposed multiscale scene encoding together with warping loss as well as window optimisation. Though the idea of each component is not novel and has been adopted in related works, there is good system contribution by extending neural dense mapping to monocular cases.
Adequate quantitative results on three public datasets show the performance gain compared to other baselines.


Weaknesses:


- As mentioned in the strength part, the unique technical novelty of this paper is a bit limited and is not as large as its system contribution.

- To my own understanding, incorporating depths not only provide direct supervision of geometry, more importantly, these geometric cues would able to largely accelerate the learning process. I am still not fully clear whether the proposed system would able to convergence in real-time as a SLAM system. More explanation from authors are appreciated. If it is real-time, I would not call it a SLAM system but a SfM/reconstruction system.

- Related to last point, in the main text, though the run time is provided, it is more desirable to show the practical tracking and mapping time efficiency (including necessary ). For example, iMAP works at 10Hz tracking and 2Hz mapping. What about the proposed system? I


- One interesting question to analysis or discuss is whether batch optimization of MLP and poses from scratch are ale to converge using proposed patch-based photometric losses. Works like BARF and NeRF-- struggles in indoor scenes. If not, could you provide more insights why incremental manner would work?

- Reconstruction results on real-world TUM, Euroc datasets or self-captured scenes are expected as only reconstructed meshes on Replica are shown. It would be interesting to see these real-world scenes captured in an less idea conditions. Some video demos would be great to see.

- The MLP decoder is fixed after intialisation stage. What is the main point of dosing so. Will there be clear performance drop enabling optimisation of MLP decoder? I am wonder whether fixing it would somehow affect the system performance to regions which are largely different to these during initialsation.


- Like iMAP, the reconstruction are mainly shown on bouded single room scenes (like Replica). Would the monocular tracking perform well on ScanNet datasets where the image capturing conditions are more challenging and the scene volume is also larger.


**Summary Of The Paper:**

This paper attempts to tackle an important topic of monocular dense mapping using neural implicit maps. Inspired by iMAP and NICE-SLAM, differential neural rendering and multi-scale feature volumes are adopted to serve as the driving force of optimisation. On important contribution claimed is the multi-scale patch-based photometric loss applied to the reference view and its overlapping views. Experiment results on three public available datasets shows the effectiveness of the proposed method in terms of tracking and mapping.

**Summary Of The Review:**

I think this is a good paper which tries to taclke an interesting and important topic.

However, I am not fully convincing about its real timeness (i.e., a SLAM) and would like to hear more from authors.
I would like to adjust my rating after my concerns are properly addressed and more information could be provided.


--------------------------------------------------------------------------------------------------------------------------------------------------------
Post Rebuttal

I have carefull read all the reviews and authors' feedbacks.
Most of my concerns have been properly addressed by the authors, and I think a monocular dense neural filed SLAM like this would be beneficial to the area. There have also been some other NeRF-SLAM system recently available, it would be good if authors could add some discussion towards these recent works.

Overall, I keep my positive rating towards this paper.

---

> ### Author Response · Authors · 2022-11-18
> **Responce to reviewer W3wz**
>
> We thank the reviewer for the valuable suggestions and address the reviewer's concerns as follows.
>
> **W1 Novelty**
>
> We agree with the reviewer that the system contribution is larger than the technical uniqueness.
>
> **W2 Runtime**
>
> It is true that additional depth input can accelerate the optimization process. Our contribution is to design an efficient method WITHOUT depth input. We have added more explanations and experiments about running time in Section 4.3 and appendix A.5. In particular, as shown in Table 5, our mapping and tracking threads take 330 ms and 72 ms respectively, faster than iMAP and slightly slower than NICE-SLAM. To achieve this runtime efficiency, we reduce the FLOPS of a single 3D sample point down to 18.76k, which is only 1/6 of that of NICE-SLAM. In this way, we can achieve near real-time performance even though our optimization involves many more samples and iterations.
>
>
> **W3 Time efficiency**
>
> Thanks for the suggestion of separating tracking and mapping. We have included that in Table 5 now. Our single-thread implementation, i.e. ours (one thread), works at 3Hz for tracking and mapping on all datasets. With two threads for tracking and mapping, i.e. ours (two threads), our method works at 15Hz tracking and 3Hz mapping. Note that in Table 7 in appendix A.5, we show the number of pixels and iterations used in the different datasets.
>
> **W4 Batch optimization of the scene and pose from scratch**
>
> Similar to NeRF-- and BARF, our method cannot optimize from scratch to solve the MLP and poses neither. Our visual SLAM setting is simpler than the general problem of `NeRF without pose`. In visual SLAM, the camera moves slowly and its pose can be initialized easily to bootstrap the optimization. Traditional direct visual SLAM methods like DSO and LSM-SLAM have similar characteristics.
>
>
> **W5 More visualization results on the real-world dataset**
>
> Thanks for the suggestion. We have included additional visualizations on real-world data in Figures 7 and 8 in the appendix A.7. We also provide an additional video demo on both synthetic and real-world datasets. Please refer to the supplementary material.
>
>
> **W6 Fixing the parameters of the decoder**
>
> As we mentioned in Section 3.1, implicit map representation suffers from a `forgetting problem`, where updating the decoder might worsen the scene map reconstructed earlier. To deal with this problem, NICE-SLAM uses a pre-trained decoder. In comparison, we learn a decoder for the current scene and fix it after initialization.
>
> We have experimented with enabling decoder optimization after initialization. The tracking accuracy is improved. But at the same time, the mapping takes more iterations to converge to enforce the geometry and color consistency throughout the entire scene.
>
> In principle, a fixed decoder might cause problems when the camera moves to a region largely different from those in initialization. We might design a mechanism to detect this problem and enable re-initialization of the decoder. But so far, we haven't met this problem in our experiments.
>
> **W7 Performance on multi-room dataset**
>
> Our method does not work very well on the ScanNet dataset. Firstly, the imaging condition there is poorer and often leads to motion blur and noisy frames. Secondly, many sequences in ScanNet include large loops, which typically require a pose graph optimization to reduce the drifting error. Such pose graph optimization is missing for neural structure-from-motion, which hinders our method to deal with camera motions with large loops.
>
>
> [1] Shrestha, R., Hu, S., Gou, M., Liu, Z., & Tan, P. (2022). A Real World Dataset for Multi-view 3D Reconstruction. ECCV 2022.

---

### Official Review · Reviewer_bVvS · 2022-10-26

**Confidence:** 4
**Clarity, Quality, Novelty And Reproducibility:** The paper is well presented and is ea…
**Correctness:** 3
**Technical Novelty And Significance:** 3
**Empirical Novelty And Significance:** 2
**Recommendation:** 6

**Strength And Weaknesses:**

Streangth
- The photometric warping loss is novel and seems to be the main source of the boosted performance compared to NICE-SLAM.
- The evaluations are thorough and consider both map and camera pose accuracies and cover both synthetic and real datasets.


Weakness
- Compared to NICE-SLAM, the main differences are (1) using different number of levels for hierarchical feature volume (2) adding the photometric warping loss. However, the camera pose accuracies in Table 1 are ~10 times better than NICE-SLAM. This is quite surprising, especially considering that the proposed method does not consume depth maps. It might be helpful to add more discussions accordingly.
- In Eq (6), why are the warped images comapred to the rendered images instead of to the real images? This looks a bit strange and more explaination should be added.
- The way of getting the visibility mask on page 5 is a bit strange. The used method does not handle occlusion at all.
- In 3.3, the naive way of finding the global keyframes are not scalable as it needs to traverse all the previous keyframes for each new frame. This will never work in a real SLAM system.
- Please add units to Table 2 and Table 3.

**Summary Of The Paper:**

This paper presents a dense SLAM method using neural implicit model as map representation. Compared to previous methods, the proposed method does not need depth maps as input. A hierarchical feature volume is used to facilitate map decoding. A photometric warping loss is proposed to be combined with the photometric rendering loss for optimizing the scene geoemtry and camera poses. Evaluation on both synthetic and real datasets show comparable or better results compared to previous methods.

**Summary Of The Review:**

Despite only moderate modifications and enhancement to NICE-SLAM, the presented work delivers surprisingly good results in some of the evaluations. I feel those parts need to be better supported by further experiments such as more dedicated ablation studies. Compared to the existing works, the overall novelty of this work is marginal. I therefore tend to reject the paper.

---

> ### Author Response · Authors · 2022-11-18
> **Response to reviewer bVvs**
>
> We thank the reviewer for the valuable suggestions and address the reviewer's concerns as follows.
>
> **W1 Tracking performance**
>
> There are mainly two reasons why our tracking accuracy is  better than NICE-SLAM:
>
> - As explained in Section 3.1, we adopt a feature volume with many hierarchies to facilitate optimization. This design choice is verified in Table 4 and Table 6 (in the appendix A.4). Reducing the number of hierarchy to 2 (like in NICE-SLAM) will generate a larger tracking error, i.e. ATE RMSE of 2.27 cm, while our feature volume of 6 hierarchies can reduce the tracking error down to 0.67 cm. The map accuracy is also improved by the deeper hierarchical feature volume.
>
> - As we discussed in the introduction, `the decoders in NICE-SLAM are pretrained, which might cause problems when generalizing to a different scene`.  A similar observation was reported in Vox-Fusion[1] too, as `NICE-SLAM uses a pre-trained geometry decoder which could reduce generalization ability`. In general, a pretrained decoder makes the camera pose harder to converge during optimization at novel scenes and thus hurts the tracking performance.
>
> **W2 Rendered image or real image in warping loss**
>
> Thanks for pointing out the mistake in our equation. It is a typo and has been corrected. We use real images to compute the warping loss.
>
> **W3 The definition of the visibility mask**
>
> It is hard to model occlusion when both cameras pose and scene depth are unknown. We have tried some commonly used occlusion masks like [2], but none of them worked well. Visual SLAM aims for real-time applications, which motivates us to use a simple approximation -- where we simply use the view cone to decide the visibility of 3D points.
>
> **W4 Keyframe selection strategy**
>
> We agree with the reviewer that a real SLAM system requires a much more sophisticated keyframe selection mechanism. This work focuses on introducing implicit map representation for dense visual SLAM. Thus, we follow a similar keyframe selection like iMAP and NICE-SLAM. More sophisticated methods might involve image retrieval or visual localization methods[2,3], which is an exciting direction for future work. We have added this discussion to the paper for better clarity.
>
> **W5 Missing unit on Table 2 and 3**
>
> We have added the unit in the caption of Tables 2 and 3.
>
>
> **W6 Limited novelty**
>
> We respect but disagree with the reviewer.
> Visual SLAM works often involve building systems, exploring different design choices, and balancing performance and efficiency. While the individual components are more or less known, it is non-trivial to design a system with the right components. As far as we know, this work presents the first dense visual SLAM with implicit map representation, which is an important step in a critical problem. Our method with only RGB input even outperforms recent works (e.g. iMAP and NICE-SLAM) with RGB-D input. We hope it will inspire future research in the community.
>
> [1] Yang, X., Li, H., Zhai, H., Ming, Y., Liu, Y., & Zhang, G. (2022). Vox-Fusion: Dense Tracking and Mapping with Voxel-based Neural Implicit Representation. arXiv preprint arXiv:2210.15858.
>
> [2] Godard, C., Mac Aodha, O., Firman, M., & Brostow, G. J. (2019). Digging into self-supervised monocular depth estimation. In Proceedings of the IEEE/CVF International Conference on Computer Vision (pp. 3828-3838).
>
> [3] Brachmann, E., Krull, A., Nowozin, S., Shotton, J., Michel, F., Gumhold, S., & Rother, C. (2017). Dsac-differentiable ransac for camera localization. In Proceedings of the IEEE conference on computer vision and pattern recognition (pp. 6684-6692).
>
> [4] Arandjelovic, R., Gronat, P., Torii, A., Pajdla, T., & Sivic, J. (2016). NetVLAD: CNN architecture for weakly supervised place recognition. In Proceedings of the IEEE conference on computer vision and pattern recognition (pp. 5297-5307).

---

### Author Response · Authors · 2022-11-18
**Response to all reviewers**

We thank all the reviewers for their insightful comments. We have revised the paper as suggested by the reviewers, and summarize the major changes as follows:

- We introduce a two-thread implementation of our method, called ours(two threads), into the paper to show the efficiency of our method. Compared with our single thread implementation, called ours(one thread), ours(two threads) has two major changes:
    - The volume size is `[64,48,32,24,16,16]cm`, while ours(one thread) contains volume size with `[64,48,32,24,16,8]cm`.
    - We divide tracking and mapping into two threads. Please refer to Appendix A.5 for more details.
    - We updated Table 1 with the result of ours(two threads).

- Table 5 is updated to reveal more details about our method's runtime and memory consumption.

- We add Table 7 to show the number of pixels and iterations used in ours and NICE-SLAM.

- More visualization results on the real-world dataset are added in Appendix A.7. Please refer to Figures 7 and 8.

- a video demo on the Replica dataset and a real-world dataset is added to supplementary material.

- We adjust some statements and add missing references in the main text.

- We highlight the text modified in the revised version.

The other concerns raised by the reviewers have also been addressed individually.

---

### Decision · Program_Chairs · 2023-01-20

**Decision:**

Accept: poster

**Justification For Why Not Higher Score:**

Despite only moderate modifications and enhancement to NICE-SLAM, the presented work delivers surprisingly good results in some of the evaluations. There have also been some other NeRF-SLAM system recently available, it would be good if authors could add some discussion towards these recent works.



**Justification For Why Not Lower Score:**

The authors have addressed the main concerns and all the reviewers agreed that the proposed monocular dense neural filed SLAM like this would be beneficial to the area.


**Metareview: Summary, Strengths And Weaknesses:**

**Summary**

This paper proposes a dense SLAM system using neural implicit model as the underlying 3D map representation. Compared to previous methods, the proposed method does not need depth maps as input. Inspired by iMAP and NICE-SLAM, differential neural rendering and multi-scale feature volumes are adopted to serve as the driving force of optimisation. Therefore, the main difference to prior work is that the proposed approach operates on RGB instead of RGB-D images, and an important contribution claimed is the multi-scale patch-based photometric loss applied to the reference view and its overlapping views. Experiment results on three public available datasets shows the effectiveness of the proposed method in terms of tracking and mapping.

**Strengths:**
* The main contribution of the paper is the proposed dense SLAM system with implicit map representation, which allows SLAM system with implicit map representation to take only RGB images as input.
* The proposed approach is technically sound and its description in the paper is easy to follow. Each part of the proposed approach is well-motivated.
* The photometric warping loss is novel and seems to be the main source of the boosted performance compared to NICE-SLAM.
* The experimental results show that the proposed approach performs well on multiple datasets and is competitive with RGB-D-based methods.

**Weaknesses:**
* Compared to NICE-SLAM, the main differences are (1) using different number of levels for hierarchical feature volume (2) adding the photometric warping loss. However, the camera pose accuracies in Table 1 are ~10 times better than NICE-SLAM.
* In 3.3, the naive way of finding the global keyframes are not scalable as it needs to traverse all the previous keyframes for each new frame. This will never work in a real SLAM system.
* Unique technical novelty of this paper is a bit limited and is not as large as its system contribution.
* One interesting question to analysis or discuss is whether batch optimization of MLP and poses from scratch are ale to converge using proposed patch-based photometric losses. Works like BARF and NeRF-- struggles in indoor scenes.
* Some design choices are not fully motivated: For the photometric warping loss, why not use the fact that the neural implicit representation also provides normal information to do a perspective warp of the patch instead of using a fronto-parallel warp? Why use SSIM and not any other robust loss (e.g., see [Park et al., Illumination change robustness in direct visual SLAM, ICRA 2017])?
* Higher memory and computational cost. The proposed method adopts hierarchical feature volumes with more levels to make the system work, which inevitably brings additional memory and time cost.


**Note From Pc:**

if the above contains the word "oral" or "spotlight" please see: "oral" presentation means -> notable-top-5% and "spotlight" means -> notable-top-25%. As stated in our emails, we are disassociating presentation type from AC recommendations